# A systematic review of behaviour change interventions to improve maternal health outcomes in sub-Saharan Africa

**Francis G. Muriithi**[1]*, **Aduragbemi Banke-Thomas**[2,3], **Gillian Forbes**[4], **Ruth W. Gakuo**[5], **Eleanor Thomas**[1], **Ioannis D. Gallos**[1,6], **Adam Devall**[1], **Arri Coomarasamy**[1], **Fabiana Lorencatto**[4]

1 WHO Collaborating Centre for Global Women's Health, Institute of Metabolism and Systems Research, University of Birmingham, Edgbaston, Birmingham, United Kingdom, 2 Department of Infectious Disease Epidemiology, London School of Hygiene and Tropical Medicine, London, United Kingdom, 3 School of Human Sciences, University of Greenwich, Old Royal Naval College, Park Row, Greenwich, London, United Kingdom, 4 Centre for Behaviour Change, University College London, London, United Kingdom, 5 Department of Nursing, Queen's Medical Centre, Nottingham University Hospitals NHS Trust, Lenton, Nottingham, United Kingdom, 6 UNDP/UNFPA/UNICEF/WHO/World Bank Special Programme of Research, Development and Research Training in Human Reproduction (HRP), Department of Sexual and Reproductive Health and Research, World Health Organization, Geneva, Switzerland

* fgm911@student.bham.ac.uk

**Data Availability Statement:** All the data relating to this manuscript are fully available and without restriction as supplementary files.

## Abstract

The rate of decline in the global burden of avoidable maternal deaths has stagnated and remains an issue of concern in many sub-Saharan Africa countries. As per the most recent evidence, an average maternal mortality ratio (MMR) of 223 deaths per 100,000 live births has been estimated globally, with sub-Saharan Africa's average MMR at 536 per 100,000 live births—more than twice the global average. Despite the high MMR, there is variation in MMR between and within sub-Saharan Africa countries. Differences in the behaviour of those accessing and/or delivering maternal healthcare may explain variations in outcomes and provide a basis for quality improvement in health systems. There is a gap in describing the landscape of interventions aimed at modifying the behaviours of those accessing and delivering maternal healthcare for improving maternal health outcomes in sub-Saharan Africa. Our objective was to extract and synthesise the target behaviours, component behaviour change strategies and outcomes of behaviour change interventions for improving maternal health outcomes in sub-Saharan Africa. We followed the Preferred Reporting Items for Systematic Reviews and Meta-Analysis (PRISMA) guidelines. Our protocol was published a priori on PROSPERO (registration number CRD42022315130). We searched ten electronic databases (PsycINFO, Cochrane Database of Systematic Reviews, International Bibliography of Social Sciences, EMBASE, MEDLINE, Scopus, CINAHL PLUS, African Index Medicus, African Journals Online, and Web of Science) and included randomised trials and quasi-experimental studies. We extracted target behaviours and specified the behavioural interventions using the Action, Actor, Context, Time, and Target (AACTT) framework. We categorised the behaviour change strategies using the intervention functions described in the Behaviour Change Wheel (BCW). We reviewed 52 articles (26 randomized trials and 26 quasi-experimental studies). They had a mixed risk of bias. Out of

**Funding:** This study was supported by the Institute of Global Innovation (IGI), University of Birmingham, as part of Dr Francis G. Muriithi's Doctoral Research Fellowship. Professor Arri Coomarasamy holds a Bill and Melinda Gates Foundation Grant. Award number INV-001393. The Institute of Global Innovation and Bill and Melinda Gates Foundation had no role in the study design, data collection and analysis, interpretation of findings, manuscript preparation, or the decision to publish.

**Competing interests:** The authors have declared that no competing interests exist.

these, 41 studies (78.8%) targeted behaviour change of those accessing maternal healthcare services, while seven studies (13.5%) focused on those delivering maternal healthcare. Four studies (7.7%) targeted mixed stakeholder groups. The studies employed a range of behaviour change strategies, including education 37 (33.3%), persuasion 20 (18%), training 19 (17.1%), enablement 16 (14.4%), environmental restructuring 8 (7.2%), modelling 6 (5.4%) and incentivisation 5 (4.5%). No studies used restriction or coercion strategies. Education was the most common strategy for changing the behaviour of those accessing maternal healthcare, while training was the most common strategy in studies targeting the behaviour of those delivering maternal healthcare. Of the 52 studies, 40 reported effective interventions, 7 were ineffective, and 5 were equivocal. A meta-analysis was not feasible due to methodological and clinical heterogeneity across the studies. In conclusion, there is evidence of effective behaviour change interventions targeted at those accessing and/or delivering maternal healthcare in sub-Saharan Africa. However, more focus should be placed on behaviour change by those delivering maternal healthcare within the health facilities to fast-track the reduction of the huge burden of avoidable maternal deaths in sub-Saharan Africa.

## Introduction

Following an average annual reduction rate (ARR) of 2.7% (UI 2.0% to 3.2%) from 2000 to 2015, the average global maternal mortality ratio (MMR) exhibited a stagnant trend between 2016 and 2020, at an ARR of—0.04% (UI -1.6% to 1.1%) [1]. A UN report estimates 287,000 global maternal deaths in 2020, with an average global MMR of 223 per 100,000 live births (UI 202 to 255) with sub-Saharan Africa's MMR at 536 per 100,000 live births (UI 469 to 640), more than double the global average [1, 2]. Moreover, in 2020, three sub-Saharan African countries exhibited MMRs surpassing 1000 deaths per 100,000 live births: South Sudan (1223; UI 746 to 2009), Chad (1063; UI 772 to 1586), and Nigeria (1047; UI 793 to 1565) [2].

Systematic review evidence from studies published between 2015 and 2020 reported that the most common causes of maternal death in sub-Saharan Africa are obstetric haemorrhage: 28.8% (95% CI = 26.5% to 31.2%), hypertensive disorders in pregnancy: 22.1% (95% CI = 19.9% to 24.2%), non-obstetric complications: 18.8% (95% CI = 16.4% to 21.2%) and pregnancy-related infections: 11.5% (95% CI = 9.8% to 13.2%) [3]. For every woman who dies during or after childbirth, approximately 20 to 30 others suffer serious injuries, infections or disabilities [4]. Their surviving neonates who become orphaned children suffer negative consequences in life, often lasting beyond one generation [5, 6].

There is wide variation in measures of maternal deaths, such as maternal mortality ratios (MMRs) between and within countries, but it is unclear what might be driving this variation [7]. Although systematic review evidence established that the percentage of skilled birth attendance and type of hospital accounted for 44% of the total variation of the hospital MMR in sub-Saharan Africa, a greater proportion (56%) remains unexplained [8]. We postulate that one explanatory factor may be differences in behaviours of those accessing and/or delivering maternal healthcare [9, 10].

The behaviour of any stakeholder may influence the delivery of maternal healthcare and outcomes either positively or negatively. Examples of stakeholders include government officials, donors, multilateral partners, civil society, the private sector, local communities, community leaders, managers, auxiliary healthcare workers, women, their partners, and their families

[11–13]. Three categories of stakeholder human behaviour may influence health outcomes, and these include behaviours that contribute to disease prevention (e.g. participation in screening programmes, behaviours that involve care-seeking and adherence to treatment (e.g. antibiotic therapy and behaviours that relate to the delivery of healthcare (e.g. evidence-based practices) [14, 15].

Within maternal health, some behaviours by those delivering care may enable improved outcomes (e.g., treating those accessing care with dignity). However, other behaviours like abuse and mistreatment of those seeking maternal healthcare services may contribute to sub-optimal maternal health outcomes [9, 15, 16].

Similarly, the behaviour of those accessing maternal healthcare may influence their engagement with healthcare advice and obstetric and general health outcomes [10, 17]. Therefore, incorporating strategies that encourage positive behaviours and evidence-based practices while discouraging negative behaviours and discontinuing potentially harmful clinical practices is essential for enhancing maternal health outcomes and addressing avoidable maternal mortality [15].

Human behaviour and behaviour change are central to the uptake of evidence-based interventions [18, 19]. We postulate that the successful implementation of the current global strategy for ending avoidable maternal mortality depends on the supportive behaviour of all the stakeholders in the maternal healthcare system [20]. The key focus of the strategy is on health system strengthening, addressing inequities in access, ensuring universal health coverage, addressing all the causes of maternal deaths and their contributing factors, and increasing country ownership, funding and sustainability–they all in one way or another require behaviour change for their effective implementation [20].

Therefore, there is a need first to identify and synthesise existing knowledge of behaviour change interventions for improving maternal health outcomes in sub-Saharan Africa. The identification and synthesis will provide a basis for describing existing interventions, identifying gaps in research and evidence that can be addressed through future research, informing current and future intervention development, and generating recommendations for policy and practice.

This systematic review aims to identify interventions to improve maternal health outcomes in sub-Saharan Africa through behaviour change and specify the behaviours and actors targeted by interventions and associated behaviour change strategies. As the stakeholders in maternal healthcare are potentially diverse, it is essential to specify whose and which behaviours have been targeted in existing interventions [12, 13]. The Action, Actor, Context, Target, and Time (AACTT) framework is a valuable tool for clarifying the behaviours of stakeholders across multiple levels of the healthcare system [21]. In addition, given that behaviour change interventions are typically complex, comprising multiple interacting components, there is a need to specify what these are as a basis for describing what has been done before and what works or does not work [22].

To facilitate this work, we adopted the following definitions of behaviour and behaviour change intervention for this systematic review:

- **Behaviour:** "Anything a person does in response to internal or external events. Actions may be overt (motor or verbal) and directly measurable or covert (activities not viewable but involving voluntary muscles) and indirectly measurable; behaviours are physical events that occur in the body and are controlled by the brain" [23].

- **Behaviour change intervention:** "Coordinated sets of activities designed to change specified behaviour patterns" [24]. Examples include education, persuasion, incentivisation, coercion, training, enablement, modelling, environmental restructuring and restrictions

We set out to answer the following research questions: 1). Which and whose behaviours are targeted by existing behaviour change interventions for improving maternal health outcomes in sub-Saharan Africa? 2). Which types of behaviour change intervention strategies are currently used for improving maternal health outcomes in sub-Saharan Africa? 3). What is the outcome of behaviour change intervention strategies for improving maternal health outcomes in sub-Saharan Africa?

## Methods

### Protocol and guidance

This systematic review followed the PRISMA (Preferred Reporting Items for Systematic Reviews and Meta-Analyses) guidelines [25]. The PRISMA checklist is presented in **S3 Table**. The protocol of this study was registered with PROSPERO (registration number CRD42022315130) [26].

### Eligibility criteria

**Inclusion criteria.** We included randomised trials and quasi-experimental studies that were published after the launch of the Safe Motherhood Initiative in 1987 and that had the following characteristics: A population was made up of actors or stakeholders whose behaviour could influence a maternal health outcome; examples include patients, partners, health workers, families, communities, managers, and policy experts; Studies whose behaviour change intervention strategies aimed at changing the behaviour of at least one of the aforementioned stakeholder groups in the context of maternal health outcomes in sub-Saharan Africa. Examples of intervention functions are as defined in the Behaviour Change Wheel. They include education, persuasion, incentivisation, coercion, training, enablement, modelling, environmental restructuring, and restrictions [24]. **See Fig 1.**

**Exclusion criteria.** We excluded articles that described behaviour (i.e. measured or assessed patterns of current practice or behaviour) or influences on behaviour (e.g. qualitative surveys or observational studies) but did not try to change the behaviour. In addition, expert opinions, commentaries, and articles that did not report the effect of the behavioural intervention were excluded.

**Search strategy.** We searched ten electronic databases systematically. These databases were PsycINFO, Cochrane Database of Systematic Reviews, International Bibliography of Social Sciences, EMBASE, MEDLINE, Scopus, CINAHL PLUS, African Index Medicus, African Journals Online, and Web of Science without any language restriction. We also conducted hand searches in key journals reporting behaviour change interventions, specifically the Annals of Behavioural Medicine, Health Psychology, Implementation Science, and Social Science and Medicine.

The search strategy included terms related to the following categories: Population (e.g., pregnant women, health care workers), concept (e.g., behaviour), behaviour change intervention, context description (e.g., an individual country name such as Kenya) and study design (e.g., trial, quasi-experimental). We used the intervention functions outlined in the Behaviour Change Wheel (BCW): Education, persuasion, incentivisation, coercion, training, enablement, modelling, environmental restructuring and restrictions to generate search terms related to different behaviour change strategies [24]. We combined the main search terms and their synonyms using the Boolean operator **"OR."** We combined the descriptor categories using the Boolean operator **"AND."** The initial search was completed on 14th March 2022 and updated on 11th August 2022. Our search strategy is in **S1 Table.**

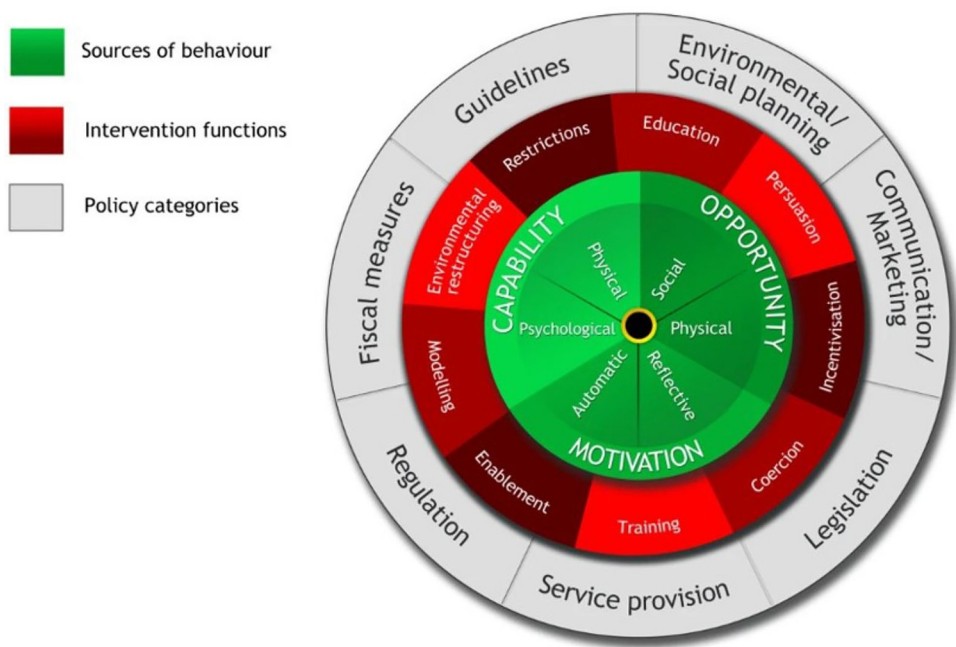

**Fig 1. Behaviour change wheel (BCW) [24].** The middle circle (shaded red) illustrates the behaviour change intervention functions.

**Article processing and study selection.** We exported all the retrieved articles into End-Note version 20 (Clarivate, Philadelphia, PA, USA), a reference management software [27]. After collation, we uploaded all the articles onto Covidence (Covidence, Melbourne, Australia), a systematic review management software for deduplication and article screening [28].

The eligibility criteria were piloted-tested by FGM and RG on a random sample of ten articles. FGM and RG independently screened the articles in two stages: title and abstract and full-text screening. Discrepancies were resolved by discussion between them or with a third reviewer.

## Data extraction

Data were extracted from the included studies by two reviewers (FGM and RG) independently using a pre-tested data extraction Google form (Google LLC, Mountain View, California, United States). Discrepancies were resolved by discussion between FGM and RG or with a third reviewer.

The data extraction categories included study characteristics (year of publication, first author, country, title, study design, aim, setting, sample size and duration), intervention descriptions, including recipients, providers and content, and study outcomes. The completed data extraction form and coding process are presented in supplementary files: **S1 and S2 Data.**

## Quality assessment

We used the Joanna Briggs critical appraisal tools for randomised controlled trials and quasi-experimental studies to assess the quality of the included studies [29]. Each tool was modified to provide a score of 1 for each domain, giving a maximum quality score of 13 for randomised controlled trials and 9 for quasi-experimental studies. Two reviewers (FGM and RG) independently assessed the quality of the included studies and agreed on a final score by consensus. We used the Robvis generic template and online software to generate a risk of bias plot [30]

## Data synthesis

To answer research question 1 regarding whose and which behaviours were targeted, intervention descriptions were coded using the Action, Actor, Context, Target, and Time (AACTT) framework for specifying target behaviours in behaviour change interventions [21, 31]. This framework was previously used in systematic reviews to specify target behaviours, such as describing interventions to improve antibiotic prescribing in long-term care facilities [32].

Two reviewers, RG and FM, independently extracted, coded and synthesised the behaviour change strategies according to the intervention functions outlined in the Behaviour Change Wheel (BCW) [24]. Conflicts were discussed and resolved by consensus between RG and FM. A behavioural scientist (GF) reviewed the extractions to check agreement with the AACTT coding and the BCW intervention function coding. We assessed the inter-coder reliability (RG vs FM) using Cohen's kappa (abstract and full-text screening process) and the percentage agreement across the five domains of the AACTT framework and the intervention function domain of the Behaviour Change Wheel (data extraction and coding).

We compared the types of intervention functions and target behaviour, target population and maternal health outcomes. In addition, we compared behaviour change strategies between those accessing and/or delivering maternal healthcare.

We explored all randomised trials and quasi-experimental studies for inclusion in a meta-analysis. However, a metanalysis was not feasible due to methodological and clinical diversity in aspects of the included studies' populations, interventions, comparisons, and outcomes (PICO).

## Deviation from protocol

We used the Joanna Briggs tools to assess the quality of randomised trials and quasi-experimental studies [29]. This deviated from the ROBINS-I tool we had proposed in our protocol (PROSPERO registration number CRD42022315130), as we anticipated that only non-randomised interventional studies would be included [26].

## Results

As outlined in the PRISMA flow chart, this systematic search yielded 18549 articles, of which 15940 remained after deduplication. A further 15286 articles were excluded during the title and abstract screening process. The full texts of the remaining 653 articles were eligible for screening. Ten full-text articles were irretrievable, and 591 were ineligible for inclusion. Fifty-two (52) articles were eligible for inclusion in the analysis. This included 26 randomised trials [33–58] and 26 quasi-experimental studies [59–84]. The article identification and selection process and output are summarised in the PRISMA Flow Chart. **See Fig 2.**

## Inter-rater reliability and quality of included studies

The inter-rater (FM and RG) reliability during the title, abstract, and full-text screening stages was calculated automatically by the Covidence systematic review management software as Cohen's kappa of 0.18597 and 0.88496, respectively. The agreement between the coders (FM and RG) across the domains of the AACTT framework was an average of 88.4%. The extent of the agreement distribution was as follows: Action 82.3%, Actor 90.4%, Context 98.1%, Target 84.6%, Time frame 98.1% and Intervention strategy 76.9%.

The quality assessment output and risk of bias chart are presented in **S2 Table** and **Fig 3**.

The risk of bias chart illustrates the assessment of the risk of bias of included articles across seven domains. Thirty had a high risk of bias, 12 were unclear, and 10 had a low risk of bias. The ROBVIS generic dataset is presented in **S3 Data.**

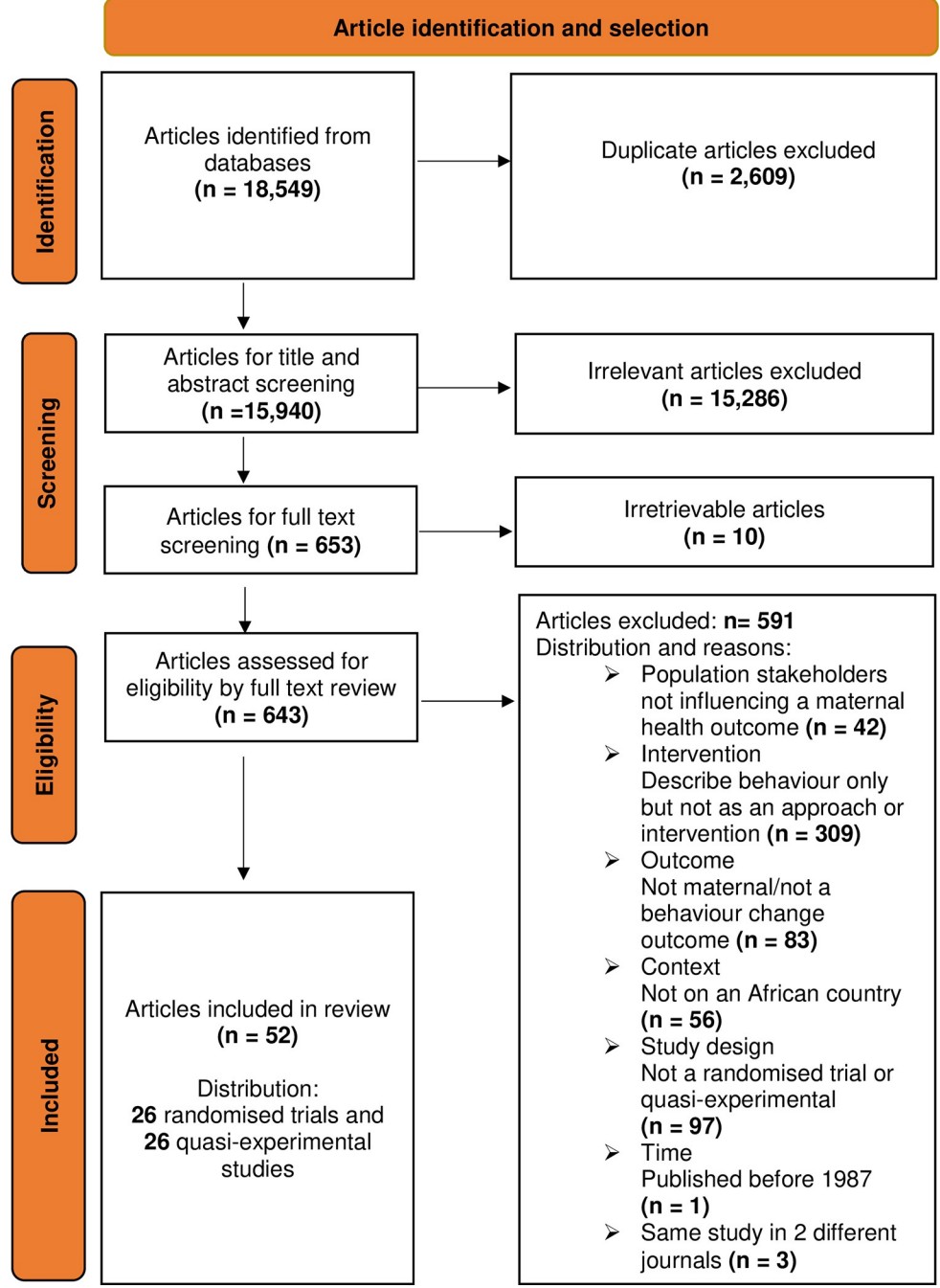

**Fig 2. PRISMA flow chart.** The flow chart illustrates *the output from the* article *identification and selection process, namely*: *Identification, title and abstract screening, and full-text screening for eligibility (both exclusion and inclusion).*

## Characteristics of included studies

Of the fifty-two (52) included studies, sixteen were conducted in Kenya [35, 36, 38, 39, 42, 45, 52, 53, 55, 56, 66, 68, 71, 76, 79, 84], seven in Ethiopia [41, 49, 54, 62, 63, 72, 78], seven in South Africa [47, 58, 60, 61, 69, 70, 77], five in Tanzania [50, 73, 74, 81, 82], four in Malawi [34, 44, 64, 80], three in Ghana [46, 65, 75], three in Nigeria [40, 43, 51], two in Egypt [59, 67] and one each in Zambia [83], Uganda [57], Zanzibar [33], Rwanda [37] and Democratic

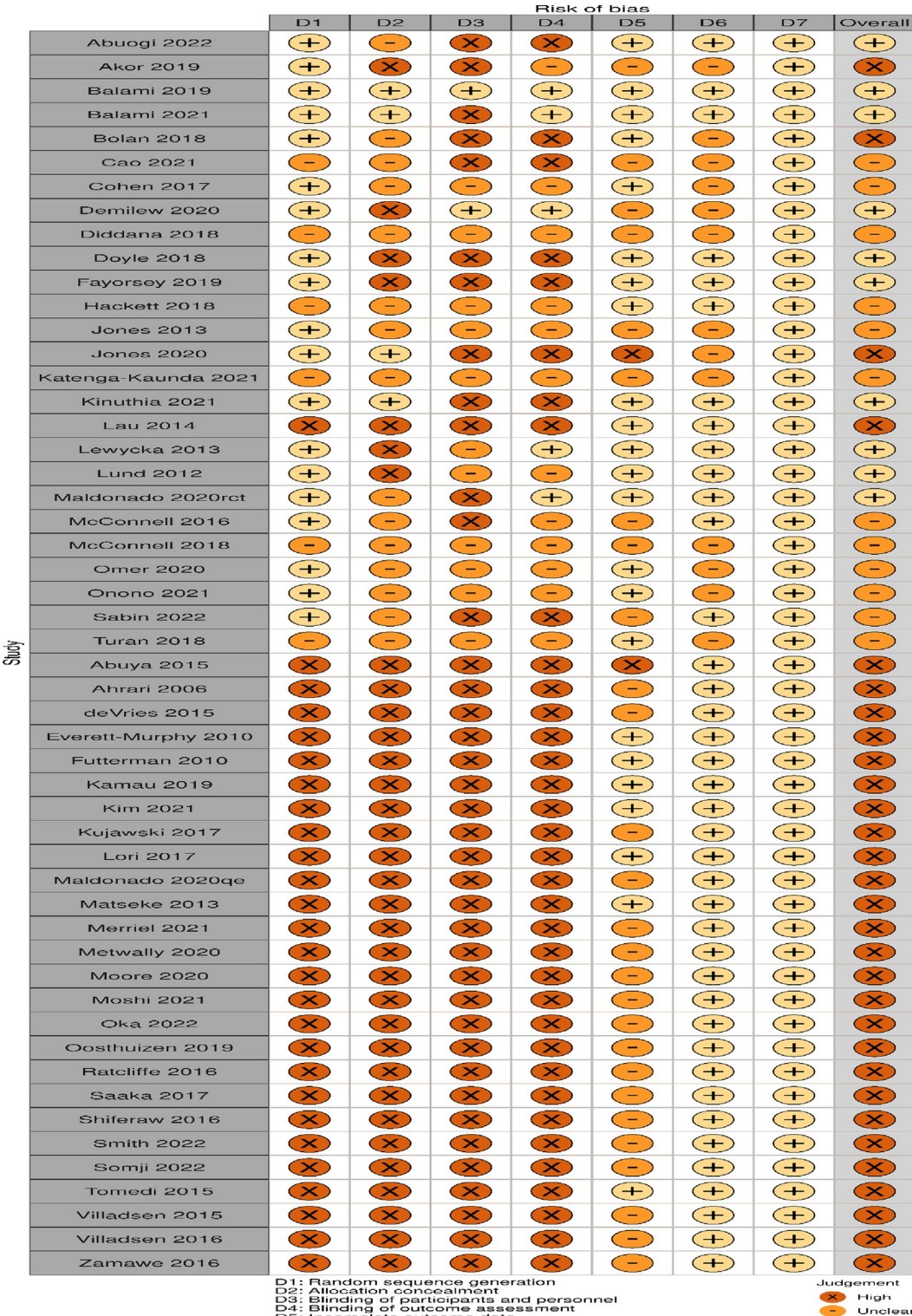

**Fig 3. Risk of bias chart.** This Robvis generated chart presents the quality assessment across seven domains summarised into three categories: "X" represents a high risk of bias, "- "represents an unclear risk of bias and "+" represents a low risk of bias.

Republic of Congo [48]. Tanzania and Zanzibar constitute one country–The United Republic of Tanzania.

Of the fifty-two included studies, 26 were randomised trials [33–58], and 26 were quasi-experimental. The sample sizes in the individual studies ranged between 13 [68] and 185,888 [34]. Behaviour change interventions in 41 studies were targeted at those accessing maternal healthcare (women and their partners) [33–47,49, 51–53, 55–61, 63–67, 69–72, 75, 76, 79, 81, 82, 84] while those delivering maternal healthcare were targeted in 7 studies [48, 50, 54, 68, 77, 78, 80]. The target population was mixed in 4 studies [62, 73, 74, 83]. Behaviour change interventions in 24 studies were informed by a behaviour change theory [37, 39, 41, 43, 45, 46, 48, 49, 51, 53–55, 58, 60, 69, 70, 73–75, 79–83]. A behaviour change theory was not mentioned in 28 studies [33–36, 38, 40, 42, 44, 47, 50, 52, 56, 57, 59, 61–68, 71, 72, 76–78, 84]. Characteristics of the included studies are presented in **S1 Data**.

### Research question (RQ) 1: Which and whose behaviours are targeted by existing behaviour change interventions for improving maternal health outcomes in sub-Saharan Africa?

In all, 41 studies [33–47, 49, 51–53, 55–61, 63–67, 69–72, 75, 76, 79, 81, 82, 84] had those accessing maternal healthcare as the target, seven studies [48, 50, 54, 68, 77, 78, 80] targeted those delivering maternal healthcare and four studies [62, 73, 74, 83] had mixed targets.

We identified and categorised various target behaviours into those accessing and/or delivering maternal healthcare and mixed target behaviours. The list of target behaviours is summarised in **Table 1**.

The target behaviours were extracted and described under the five domains of the Action, Actor, Context, Target, Time (AACTT) framework [21]. The action was specified in 52 (100%)

**Table 1. Summary table of target behaviours.**

| Category | Target behaviour | Count | Citations |
|---|---|---|---|
| Self-care behaviour 25 studies (48%) | Adoption of healthy behaviours and birth preparedness | 11 | [39, 53, 63–66, 72, 75, 81, 82, 84] |
| | Dietary and lifestyle behaviours | 4 | [41, 44, 49, 59] |
| | Intimate partner violence deterrence behaviours | 3 | [40, 46, 61] |
| | Malaria prevention behaviours | 2 | [43, 51] |
| | Drug use and misuse cessation behaviours (alcohol, smoking) | 2 | [60, 70] |
| | Sexual risk behaviours | 2 | [47, 76] |
| | Maternal depression prevention | 1 | [77] |
| Healthcare seeking behaviour 16 studies (31%) | Care seeking behaviours | 7 | [34–36, 42, 58, 67, 71] |
| | Adherence behaviours | 7 | [38, 45, 52, 55–57, 69] |
| | Male involvement | 1 | [37] |
| | Care utilisation | 1 | [33] |
| Healthcare delivery behaviour 7 studies (13%) | Respectful maternity care | 2 | [68, 77] |
| | Managing obstetric emergencies | 1 | [48] |
| | Care utilisation | 1 | [50] |
| | Professional workplace relationships | 1 | [80] |
| | Teamwork | 1 | [78] |
| | Promotion of healthy behaviours | 1 | [54] |
| Mixed, i.e., both healthcare-seeking and healthcare-delivery behaviours 4 studies (8%) | Respectful maternity care | 4 | [62, 73, 74, 83] |

studies, the actor in 49 (94%) studies, the context in 52 (100%) studies, the target in 52 (100%) studies and the timeframe in 52 (100%) studies. The results of this synthesis are presented in **S2 Data** and **Table 2**.

### Research question (RQ) 2: Which types of behaviour change intervention strategies are currently used for improving maternal health outcomes in sub-Saharan Africa?

We identified various behaviour change intervention strategies used either singly or in combination. The scope of behaviour change strategies we identified is tabulated in **Table 3** and ranked by frequency in **Fig 4**.

In studies targeting behaviour change of stakeholders accessing maternal healthcare, education was the most frequently employed strategy, while training was most commonly utilized in studies targeting behaviour change of stakeholders delivering maternal healthcare. However, there was an overlap in the choice of behaviour change strategies between the two stakeholder groups.

A comparison of behaviour change strategies by category of target stakeholder is presented in **Fig 5**.

### Research question (RQ) 3: What is the outcome of behaviour change intervention strategies for improving maternal health outcomes in sub-Saharan Africa?

The behaviour change intervention strategies were reported as effective in 40 studies, ineffective in 7 and equivocal in 5.

Effective behaviour change interventions amongst those delivering maternal healthcare targeted the following behaviours: Managing disrespectful and abusive behaviours [68, 77], managing obstetric emergencies confidently [48], community healthcare worker professional behaviour [50], respectful behaviour amongst professional colleagues [80], teamwork and communication behaviours [78] and supporting women to adopt healthy behaviours in antenatal care [54].

Amongst those accessing maternal healthcare, effective behaviour change interventions targeted the following behaviours: Healthy nutrition and dietary behaviours [41, 44, 49, 59], coping with and deterrence against intimate partner violence [37, 40, 61], malaria prevention behaviours [43, 51], smoking cessation behaviour [60], retention and adherence to Human Immunodeficiency Virus (HIV) care behaviour [52, 56], HIV infection risk reduction behaviour [39, 47], care-seeking behaviour [34, 35, 42, 67, 71], anaemia prevention behaviour [76], birth preparedness and complication readiness behaviour [65, 75, 81, 82], attendance for skilled delivery behaviour [33, 53, 66], postpartum contraception take up behaviour [38] and behaviours that support effective utilisation of maternal healthcare services [64].

Effective behaviour change interventions with mixed targets (both those accessing and delivering maternal health care) targeted two behaviours: Promotion of respectful maternity care [73, 74, 83] and partner support during and after pregnancy and malaria prevention behaviour in maternal healthcare delivery [62].

Ineffective behaviour change interventions all targeted those accessing maternal healthcare. They included HIV postpartum adherence to treatment [45, 55, 57], intimate partner violence [46], maternal depression [79] and adoption of healthy behaviours [72, 84]. It is not clear why these interventions were ineffective.

Equivocal outcomes following behaviour change interventions were reported for maternal healthcare seeking [36, 58], adoption of healthy behaviours [63], adherence to the prevention

**Table 2. Description of target behaviours and behaviour change intervention strategies.**

| Article (First author, year) | Action (observable or measurable behaviour) | Actor (does or could do the action) | Context (location, emotional context, or social setting) | Target (person/people) | Time frame (when the action is performed) | Intervention strategy | Intervention Description | Measured outcomes |
|---|---|---|---|---|---|---|---|---|
| Abuogi 2022 | Healthcare-seeking behaviour (retention in care and adherence to treatment) | Healthcare workers and community mentor mothers (cMMs) | Randomisation at the facility level, intervention at the individual patient level | Women with HIV | Antenatal | Education, enablement, and persuasion. | Participants received free tailored text messages delivered by automated texting software. Message content focused on medication and clinic adherence without explicit mention of HIV or ART, as well as promotion of maternal and child healthcare services timed to the stage of pregnancy and age of the infant post-delivery. Community mentor mothers (cMMs) engage with an HIV-infected pregnant woman at earliest identification within the antenatal clinic and follow these women up to at least 1-year postpartum, offering Prevention of Mother to Child Transmission) PMTCT Peer Education and Psychosocial Support. | Self-reported adherence to Anti-Retroviral Therapy (ART). Viral suppression by serial viral load measurement. Infant outcome of HIV test status at 12 months. |
| Abuya 2015 | Healthcare delivery behaviour (Managing disrespectful and abusive behaviours') | Trained nurses and midwives | Health facility and community | Care providers | Intrapartum | Training and persuasion | Working with policymakers to encourage greater focus on disrespect and abuse (D & A), training providers on respectful maternity care, and strengthening linkages between the facility and community for accountability and governance. | How patients were treated, and what they were told. Percentage of women responding to the question: were you treated in a way that made you feel humiliated or disrespected? On a Likert scale. A yes or no response to questions assessing the effect of the intervention on the six categories of D & A, including the occurrence of physical abuse, violation of privacy as well as confidentiality, verbal abuse, detainment, and abandonment. |
| Ahrari 2006 | Self-care behaviour (Adopting healthy behaviours) | Trained volunteers, positively deviant mothers, and their mothers-in-law | Community | Women at risk of delivering low birth-weight infants | Antenatal | Education, enablement, and persuasion | Social mobilisation and training: Weekly IMPRESS (improved pregnancy through education and supplementation) sessions of food supplementation and counselling. Community health workers visited each IMPRESS attendee at home once a month to review messages, solve problems, and encourage practices not yet adopted. | Birthweights of all newborns within 48 hours of birth. Bi-monthly weight and length measurements. Self-reported pregnancy-related behavioural data compared intake of food or rest during pregnancy with the practice prior to pregnancy. |
| Akor 2019 | Self-care behaviour (Managing abusive behaviours) | Counsellors | Health facility (antenatal clinic) | Pregnant intimate partner violence victims | Antenatal | Enablement and training | Counselling was done using the SOS-DoC framework (S—offer support and assess safety; O—discuss options; S—validate patient's strengths; Do—document observations, assessment, and plans; C—offer continuity). A combination of two therapeutic counselling techniques,20 i.e., Nondirective counselling, which aims at encouraging the client to discuss her/his problems with the counsellor who, through listening, affirms the patient's worth and allows her/him to take time to express their thoughts; and problem-solving therapy, which involves systematically teaching generic skills in active problem solving to reduce stress and enhance self-efficacy. | Abuse Assessment Scale (AAS) for recruitment of participants The family function was assessed using the SCORE-15 (systemic clinical outcome and routine evaluation) https://www.aft.org.uk/page/score |
| Balami 2019 | Self-care behaviour (Malaria preventive behaviours) | Facilitator (unspecified) | Antenatal clinic | Antenatal attendees | Antenatal | Education | 4-h health education on malaria, guided by a module developed based on the IMB theory, while the control group received health education on breastfeeding for a similar duration and by the same facilitator. | Knowledge of malaria transmission. Personal motivation. Social motivation. Behavioural skills |

*(Continued)*

**Table 2.** (Continued)

| Article (First author, year) | Action (observable or measurable behaviour) | Actor (does or could do the action) | Context (location, emotional context, or social setting) | Target (person/people) | Time frame (when the action is performed) | Intervention strategy | Intervention Description | Measured outcomes |
|---|---|---|---|---|---|---|---|---|
| Balami 2021 | Self-care behaviour (Malaria preventive behaviours) | Midwife | Health facility | Pregnant women | Antenatal | Education | A four-hour malaria health education intervention in the Hausa language, which was developed based on the IMB model, while the control group received a similarly designed health education on breastfeeding. | Reported Insecticide Treated Network (ITN) use, Reported Intermittent Preventive Treatment in pregnancy (IPTp) uptake, reported malaria diagnosis, haematocrit, pregnancy outcome, and babies' birth weights. |
| Bolan 2018 | Healthcare delivery behaviour (Managing basic obstetric emergencies) with confidence | Ministry of health supervisors and external trainers | Health facility | Medical doctors (MDs), nurses, and midwives | Intrapartum, postnatal | Training, education | Safe delivery app (SDA) training tool and job aid—Designed to reinforce the capability and confidence of healthcare workers in low-income countries on how to manage basic obstetric and neonatal emergencies. The content of the app is based on global clinical Basic Emergency Obstetric Care (BEmONC) guidelines. The SDA conveys knowledge and skills via animated videos and instructions on key procedures. It also contains information on essential drugs for BEmONC. | Self-confidence scores for 12 essential BEmONC services. Knowledge scores 2 key BEmONC services (management of postpartum haemorrhage and neonatal resuscitation) |
| Cao 2021 | Self-care behaviour (Managing violent and abusive behaviour) | Lay counsellors supervised by community health officers and NGO field staff | Community | Pregnant women and their partners | Antenatal | Education and enablement | Maternal mental health intervention called Integrated Mothers and Babies Course & Early Childhood Development (iMBC/ECD) and social support and/or couple communication. | Depression score on locally validated 9-item Patient Health Questionnaire (PHQ-9). Controlling behaviours, emotional violence, physical violence as well as sexual violence were assessed by the items in the Ghana Demographic and Health Survey. Social support was measured using The Modified Medical Outcomes Study Social Support Survey (mMOS-SS). Couple communication was measured by the communication domain from the Couple Functionality Assessment (CFAT). |
| Cohen 2017 | Healthcare seeking behaviour | Unclear | Community | Pregnant women | Antenatal | Incentivisation | Women received Labelled Cash Transfer (LCT) plus an additional cash transfer if they delivered in a facility to which they had committed during pregnancy. | Whether the mother delivered at the facility she most wanted and perceived as having the highest quality, and whether the cash transfers influenced the mode of transport or distance travelled to the facility. Women's perceptions of the quality of care during delivery. Mother's perception of the availability of drugs, supplies, and equipment; knowledge and competence of the health care workers; and cleanliness of the facility. Measures of patient-perceived non-technical quality by any experience of disrespect or abuse, and Likert scales for the communication skills, friendliness, and respectfulness of health care workers. Whether the delivery facility met the standards for care in three domains—routine, basic emergency, and comprehensive emergency care—for both obstetric and newborn care. |

*(Continued)*

**Table 2.** (Continued)

| Article (First author, year) | Action (observable or measurable behaviour) | Actor (does or could do the action) | Context (location, emotional context, or social setting) | Target (person/people) | Time frame (when the action is performed) | Intervention strategy | Intervention Description | Measured outcomes |
|---|---|---|---|---|---|---|---|---|
| Demilew 2020 | Self-care behaviour (Nutritional/dietary behaviour) | Trained counsellors | Community | Pregnant women | Antenatal | Enablement, education, and persuasion | The core contents of the counselling guide were: Increasing meal frequency and portion size with increasing gestational age; taking diversified meals by giving emphasis to iron-rich foods, animal products, fruits, and vegetables; the consumption of iron/folic acid supplements and iodised salt; reducing heavy workload, taking day rest, impregnated bed net use and utilisation of health care services; the consequences of taking inadequate nutrient, susceptibility to and severity of the consequences of insufficient nutrient intakes, benefits of taking an adequate amount of diversified meals and barriers that interfere with taking a balanced diet; Attitude, subjective norms, self-efficacy, perceived control, intention, knowledge, and dietary practice were assessed during each counselling session. Individual nutrition counselling was given through a home visit. | Dietary practice assessed using a food frequency questionnaire (FFQ). |
| deVries 2015 | Self-care behaviour (Abstinence or reduction of alcohol consumption behaviour) | Field staff (social workers and nurses) | Community health clinics | Pregnant women | Antenatal | Education and training | A set of social service functions focused on education, coaching, and support to help women assess their inner strengths and external resources to change their alcohol drinking behaviour. | Rates of alcohol consumption (quantity, frequency, timing, and context of drinking, such as the number of drinks consumed per day, per week, and on weekends) captured with a seven-day recall measurement of alcohol consumption at baseline and 6 monthly visits. |
| Diddana 2018 | Self-care behaviour (Dietary behaviours) | Trained community health volunteers/health extension workers | Community | Pregnant women | Antenatal | Education | Nutrition education intervention was given to pregnant women: (1) susceptibility of the pregnant women and foetus to malnutrition due to inappropriate dietary practices, nutrient deficiency, or over-nutrient intake; (2) severity of malnutrition such as wasting/thinness and overweight/obesity and high risk of foetus to intrauterine growth retardation, brain development, and cognitive function due to macro- and micronutrient deficiency; (3) benefits of right eating or dietary practices on women nutritional status and foetus health, (4) barriers to practice appropriate good dietary practices; and (5) self-confidence/efficacy to follow right. dietary practices. The education was provided using theoretical session, poster, brochures, flipchart, and whiteboard. | Nutritional knowledge. Health Belief Model constructs. |
| Doyle 2018 | Self-care behaviour (Partner support during pregnancy and managing partner relationship behaviour) | Community volunteers (local fathers), local nurses and police officers | Community at local schools and administrative offices | Pregnant women and their partners | Antenatal | Education, training, persuasion, and modelling | The Bandebereho intervention. The Bandebereho couples' intervention engaged men and their partners in participatory, small-group sessions of critical reflection and dialogue. 15 Sessions addressed: Gender and power; fatherhood; couple communication and decision-making; IPV; caregiving; child development; and male engagement in reproductive and maternal health. The MenCare+ program was known as Bandebereho, or "role model," as it aimed to transform norms around masculinity by demonstrating positive models of fatherhood. | Reproductive and maternal health behaviours, including men's participation in antenatal care (ANC) visits. Women's experiences of Intimate Partner Violence (IPV) Use of physical punishment against children. Gendered division of childcare and household tasks. Men's dominance in household decision-making |
| Everett-Murphy 2010 | Self-care behaviour (Smoking cessation behaviour) | Midwives and peer counsellors | Health facility | Pregnant women | Antenatal | Education, enablement, and persuasion | The intervention cohort was offered self-help quit materials in the context of brief counselling by midwives and peer counsellors. | Quit rates measured by urinary cotinine towards the end of pregnancy (36–39 weeks gestation). |

*(Continued)*

**Table 2.** (Continued)

| Article (First author, year) | Action (observable or measurable behaviour) | Actor (does or could do the action) | Context (location, emotional context, or social setting) | Target (person/people) | Time frame (when the action is performed) | Intervention strategy | Intervention Description | Measured outcomes |
|---|---|---|---|---|---|---|---|---|
| Fayorsey 2019 | Healthcare seeking behaviour | Trained and mentored lay counsellors "Mama mshauri" | Health facility | HIV-positive pregnant women | Antenatal | Education and persuasion | In addition to Standard of Care (SOC) services, participants who were randomised to the intervention arm were assigned a lay counsellor, called a "Mama Mshauri," at enrolment. The Mama Mshauri provided the following: (1) individualised PMTCT health education using a standardised flip chart during home and clinic visits; (2) retention and adherence support; (3) phone and SMS appointment reminders; (4) and follow-up and tracking for missed clinic visits. Mama Mshauri assisted with expediting service provision, enhancing communication between participants and health providers, assisting participants to identify and problem-solve barriers to retention and adherence, and providing psychosocial support and counselling. | Mother–infant attrition i.e. the proportion of mother–infant pairs not retained in the clinic at 6 months postpartum because of mother or infant death or lost to follow-up (LTFU). Maternal clinic attendance Other outcome measures: Maternal viral suppression at 6 months postpartum, proportion retained and virally suppresses at 6. months postpartum, and measurements of PMTCT service uptake, exclusive breastfeeding, and infant HIV testing at 6 weeks and 6 months. |
| Futterman 2010 | Self-care behaviour (Well-being seeking behaviours) | Mentor mothers | Health facility | Pregnant women | Antenatal | Education and training | Women at the intervention site (Gugulethu) received the support of a mentor mother and attended an eight-session Mamekhaya CBI. The eight sessions focused on four broad topics: 1. Healthy Living (staying in care, dealing with symptoms, learning about HIV and when to take ARVs, family planning, and condom use); 2. Feeling Happy and Strong (disclosure, dealing with stigma, finding support, feeling hope, avoiding negative emotions, dealing with domestic violence and substance abuse); 3. Partnering and Preventing Transmission (infant feeding practices, general HIV precautions, partner testing, disclosure, safer sex); and 4. Parenting (feeding choice, immunisation of the baby, adherence to pre- and postnatal baby treatment, testing the baby, planning custody, forming an attachment to the baby). | Self-reports of adherence to Prevention of Mother to Child transmission of HIV (PMTCT) practices. |
| Hackett 2018 | Healthcare delivery behaviour (Community health worker healthcare delivery practices and behaviours) | World Vision (unspecified) | Community | Community health workers | Antenatal | Persuasion and enablement | Smartphone intervention aimed at increasing women's demand for, and utilisation of facility delivery. During prenatal household visits, the application guides CHW through electronic "decision tree" protocols, directing them to specific health counselling topics and messages based on the woman's gestational age, and her answers to various diagnostic questions. The tool directs CHW to lessons in the photo book and reminds them to counsel women on the benefits of seeking antenatal care, developing a birth plan, and seeking skilled birth assistance at health facilities. The application also assists with danger sign identification, flags clients who require immediate referral to health facilities, and reminds CHW to follow up with clients previously referred. | Postnatal report of delivery at, or while in transit to, a health facility captured during postnatal household surveys. |

*(Continued)*

**Table 2.** (Continued)

| Article (First author, year) | Action (observable or measurable behaviour) | Actor (does or could do the action) | Context (location, emotional context, or social setting) | Target (person/people) | Time frame (when the action is performed) | Intervention strategy | Intervention Description | Measured outcomes |
|---|---|---|---|---|---|---|---|---|
| Jones 2013 | Self-care behaviour (Sexual risk behaviour) | Gender-matched trained HIV counsellors | Health facility | Pregnant women and their partners | Antenatal | Training and enablement | The PartnerPlus intervention combined key elements of two evidence-based interventions. (The Partner Project: a couples' behavioural HIV risk reduction intervention plus an intervention designed to enhance PMTCT uptake) into a comprehensive couples based PMTCT intervention. The intervention consisted of four weekly 90-to-120-minute sessions emphasising cognitive-behavioural skill building to improve communication, sexual negotiation, conflict resolution, STI/HIV prevention, PMTCT, use of male and female condoms, and gender-relevant issues. | HIV serostatus and partner disclosure Frequency of multiple partnering and sexual barrier use with nonprimary partners over the past month using a sexual activities questionnaire adapted from the Sexual Risk Behaviour Assessment Schedule (SERBAS). Sexual activities each day of the previous week and use of condoms using a sexual activity diary. Conflict resolution strategies using negotiation, verbal aggression or physical violence using a modified version of the Conflict Tactics Scale. Knowledge concerning HIV transmission, PMTCT and AIDS was assessed using 13 items adapted from an AIDS-related knowledge scale. |
| Jones 2020 | Healthcare seeking behaviour | Obstetrician and research assistant | Health facility and community | Postnatal women | Postnatal | Education and persuasion | Each of the three intervention arms had a different combination of the SMS content. Arm 2 received the "postpartum checklist" (PPC) the week following discharge, which consisted of Yes/No questions regarding their postpartum state. Arm 3 received the "postpartum checklist" plus general postnatal care messages and reminders in the 4 weeks after discharge (PPC +PNC). Arm 4 received the "postpartum checklist" plus family planning messages/reminders between 4 and 6 weeks after discharge (PPC +FP). | Danger sign knowledge and care seeking related to danger signs. General postnatal care (immunization, mother's postpartum checkup, and child wellness visits). Family planning (uptake of family planning, planning to uptake family planning, LARC method use). |
| Kamau 2019 | Self-care behaviour (Sexual risk behaviour) | Gender-matched HIV counsellors | Health facility | Pregnant women and their partners | Antenatal | Education | Iron and Folic Acid Supplementation (IFAS) health education with weekly supplements and follow-ups to pregnant women in the intervention group | Level of knowledge about Iron and Folic Acid Supplementation (IFAS) during pregnancy. Respondents' attitude towards IFAS. |
| Katenga-Kaunda 2021 | Self-care behaviour (Nutrition behaviour) | Trained lay counsellors | Community | Pregnant women | Antenatal | Education, enablement, and modelling | Nutrition education, cooking demonstrations and home-based counselling. | Changes in nutrition knowledge and dietary diversity |
| Kim 2021 | Self-care behaviour (Maternal depression self-awareness) | Lay counsellors and community health volunteers | Health facility and community (home) | Pregnant women | Antenatal | Education and persuasion | An intervention called the Integrated Mothers and Babies Course & Early Childhood Development (iMBC/ECD) program. The intervention groups received iMBC/ECD content implemented over 14 in-person, group-based sessions spanning a period of 7 months (meetings occurring every two weeks). After completing the formal sessions, five iMBC/ECD follow-up booster sessions were conducted over the span of the next six months for a refresher on lessons learned during the program and to aid in practicing learned skills. Lead mothers conducted home visits to reinforce education messages and reported group and home visit attendance to their local CHV The control groups received the same early childhood development education content during regular biweekly care group meetings as a part of the ongoing SCORE-ECD program. For both the intervention and control groups, care groups continued to meet beyond the duration of the study. | Maternal depression was assessed using the Patient Health Questionnaire (PHQ-9) which has been previously validated in Kenya. Children's social and emotional development was measured using the Ages and Stages Questionnaires: Social-Emotional, Second Edition (ASQ:SE-2). |

*(Continued)*

**Table 2.** (Continued)

| Article (First author, year) | Action (observable or measurable behaviour) | Actor (does or could do the action) | Context (location, emotional context, or social setting) | Target (person/people) | Time frame (when the action is performed) | Intervention strategy | Intervention Description | Measured outcomes |
|---|---|---|---|---|---|---|---|---|
| Kinuthia 2021 | Healthcare seeking behaviour | Study nurses | Health facility | Pregnant and postpartum women | Antenatal, postnatal | Persuasion | Intervention participants received visit reminders and prespecified weekly SMS on antiretroviral therapy (ART) adherence and MCH, tailored to their characteristics and timing. | Primary trial outcomes were maternal virologic no suppression, on-time visit attendance, loss to follow-up, and infant HIV infection or death. Secondary outcomes were maternal ART adherence and ART resistance. |
| Kujawski 2017 | Healthcare delivery behaviour (Using respectful maternity care) behaviours | Local community and health system stakeholders | Health facility and community | Local community and health system stakeholders | Postpartum | Environmental restructuring | The intervention consisted of a client service charter and a facility-based, quality-improvement process aimed to redefine norms and practices for respectful maternity care. | Self-reported experience of disrespectful or abusive treatment during labour and delivery. Secondary outcomes included affirmative responses for each of the questions in the categories of disrespect and abuse. |
| Lau 2014 | Healthcare seeking behaviour | Obstetricians, midwives, and health promoters | Health facility | Pregnant women | Antenatal | Education | SMS reminders and health information tailored by trimester. | Antenatal Knowledge and health-related behaviour |
| Lewycka 2013 | Healthcare seeking behaviour | Women's groups and trained volunteer peer counsellors | Community | Pregnant women | Antenatal, postnatal | Education, enablement, and modelling | Volunteer peer counsellors provided health education about exclusive breastfeeding, infant care, immunisations, prevention of MTCT (PMTCT), and family planning. They also supported women with breast problems and raised awareness of timely care-seeking. Counsellors used an intervention manual describing visit content, and a simple picture book (adapted from manuals published by WHO, Save the Children, and Linkages. | Primary outcomes were maternal, perinatal, neonatal, and infant mortality rates (MMR, PMR, NMR, and IMR, respectively) for the clusters assigned to the women's group intervention. Infant Mortality Rate (IMR) and rates of exclusive breastfeeding (EBF) in the first 6 months for those assigned to the volunteer peer counselling intervention. Secondary outcomes for the women's group intervention were maternal and infant morbidity, skilled antenatal, delivery, and postnatal care, tetanus toxoid immunisation, use of malaria prophylaxis, insecticide-treated bed nets during pregnancy, and PMTCT services, infant immunisations, early EBF, and reduced use of pre-lacteal feeds. For the volunteer peer counselling intervention, neonatal mortality and infant morbidity rates were secondary outcomes, and caretaker practices included duration of EBF, time to initiating breastfeeding, use of pre-lacteal feeds, time to weaning, management of breast problems, and family planning uptake, including condom use. |
| Lori 2017 | Self-care behaviour (Healthy behaviours during pregnancy) | Skilled care provider | Health facility | Pregnant women | Antenatal | Education | The teaching component for women in individual care consisted of the midwife providing information in a lecture format to all women who presented for care that day on standard ANC educational content (i.e., danger signs, breastfeeding, birth preparedness and complication readiness, etc.) prior to their individual appointment with the midwife. The same educational content was presented as a facilitated discussion in the intervention group. | Knowledge gained by pregnant women during their experiences with antenatal care. Self-care knowledge, birth preparedness, complication readiness, breastfeeding knowledge, and postpartum danger signs. |

*(Continued)*

**Table 2.** (Continued)

| Article (First author, year) | Action (observable or measurable behaviour) | Actor (does or could do the action) | Context (location, emotional context, or social setting) | Target (person/ people) | Time frame (when the action is performed) | Intervention strategy | Intervention Description | Measured outcomes |
|---|---|---|---|---|---|---|---|---|
| Lund 2012 | Healthcare-seeking behaviours | Trained research assistants | Health facility | Pregnant women | Antenatal, postnatal | Education, persuasion, and environmental restructuring | The wired mothers' intervention consisted of two components: an automated short messaging service (SMS) system providing wired mothers with unidirectional text messaging, and a mobile phone voucher system providing the possibility for direct two-way communication between wired mothers and their primary healthcare providers. Although only women with registered phone numbers received text messages, all women in the intervention group were given mobile phone vouchers to contact their local primary healthcare provider in the case of an emergency. The wired mothers were not provided with mobile phones or access to power supplies for mobile phone charging because researchers wished to study a realistic environment that could be replicable at scale and in other settings. To improve the communication and referral mechanisms between different levels of care, providers at primary healthcare facilities and hospitals were given mobile phone. The aim of the SMS component was to provide simple health education and appointment reminders to encourage attendance to routine antenatal care, skilled delivery attendance and postnatal care. The mobile phone vouchers allowed all wired mothers to communicate directly with primary healthcare providers and to access emergency obstetric care through improved communication and referral links from primary healthcare facilities to hospitals | Skilled delivery attendance. |
| Maldonado 2020 | Self-care behaviour (Healthy behaviours during pregnancy and puerperium) | Community health volunteers supervised by Community health extension workers | Health facility and community | Pregnant women | Antenatal, postnatal | Education and persuasion | Intervention clusters were invited to bimonthly, group-based, CHV-led health lessons (Chamas); control clusters had monthly, individual CHV home visits (standard of care). Group-based, CHV-led health education programme that supports women during the first 1000 days of their child's life. | Primary outcome was the odds of facility-based Delivery. Secondary MNCH outcomes included: The relative proportion of women who attended at least four ANC visits, received a CHV home-visit within 48 h postpartum, EBF to 6 months, adopted a modern FP method, and adopted a long-term or permanent FP method. |
| Maldonado 2020 | Healthcare-seeking behaviours | Community health volunteers (CHV) | Health facility and community | Pregnant women | Antenatal, postnatal | Training, enablement and incentivisation. | Women attending Chamas convened twice per month for 12 months to attend a total of 24 CHV-facilitated group health education and microfinance sessions. Each session consisted of a 60 to 90-min participatory lesson on one health (i.e., antenatal care, family planning) and one social (i.e., intimate partner violence, microfinance literacy). Upon joining the program, women agreed to practice key MNCH behaviours, namely, to deliver in a health facility, attend at least four ANC visits, EBF to six months, receive a CHV home visit within 48 h of delivery, consider a long-term method of FP, ensure their infant received OPV0, and save money to finance health expenditures. The group provided loans that amounted to a multiple of the individual member's savings and returned a dividend payment based on interest accrued at the end of the year | Facility-based delivery at 12-month follow up. Secondary outcomes included: attending adequate ANC (defined as attending at least four visits per Republic of Kenya MOH guidelines), receiving a 48-hour postpartum home visit, exclusively breastfeeding for 6months, adopting a modern contraceptive method, immunising infants with the oral polio vaccine within 2weeks postpartum, immunising infants with the measles vaccine (measles I) by 12 months of age and completing the infant immunisation series per WHO and Republic of Kenya MOH standards by 12 months of age. Microfinance data and perceived levels of peer support and financial empowerment. |

*(Continued)*

**Table 2.** (Continued)

| Article (First author, year) | Action (observable or measurable behaviour) | Actor (does or could do the action) | Context (location, emotional context, or social setting) | Target (person/people) | Time frame (when the action is performed) | Intervention strategy | Intervention Description | Measured outcomes |
|---|---|---|---|---|---|---|---|---|
| Matseke 2013 | Self-care behaviour (Managing abusive and violent behaviour).' | Trained community workers | Health facility | Pregnant women | Antenatal | Training | A 20-minute session that included: a) Supportive care. The community worker serves as an available, interested, and empathic listener. Women are encouraged to discuss the violence they experience, their life situations, and the issues they face. b) Anticipatory guidance. Women are told what to expect if they decide to access legal aid, law enforcement, shelter, or counselling services, as well as the risks associated with leaving the abuser, having the abuser arrested, or applying for a protection order. c) Guided referrals. The community worker offers referrals tailored to the individual woman's needs (e.g., legal aid, shelter, counselling services, etc.). | Potential risk of becoming a victim of femicide using the danger assessment scale. |
| McConnell 2016 | Healthcare seeking behaviours | Community Health workers | Community | Postnatal women | Postnatal | Training | For women randomly assigned to the checklist groups, CHWs were trained to screen for maternal and newborn danger signs, to deliver targeted postnatal health education, and to refer mothers and their newborns to facility-based care, if necessary, using a checklist. | Care-seeking behaviours for mother and newborns and both knowledge and practice of infant care, nutrition, feeding and recognition of danger signs. |
| McConnell 2018 | Healthcare-seeking behaviour (appointment attendance and medication adherence) | Trained enumerators | Community | Pregnant and Postnatal women | Antenatal, postnatal | Incentivisation | Both voucher arms received a voucher for free contraceptive methods to be redeemed at a Jacaranda Health facility | Primary outcome was self-reported current use of a modern contraceptive method. |
| Merriel 2021 | Healthcare delivery behaviour (Respectful behaviour amongst colleagues) | Facilitators | Health facility | Maternity health workers | Antenatal, intrapartum, postnatal | Enablement | Motivational intervention in the form of appreciative inquiry. | Basic psychological needs, job satisfaction and work-related quality of life |
| Metwally 2020 | Healthcare seeking behaviours | Trained health workers | Health facility and Community | Pregnant women | Antenatal, intrapartum, postnatal | Education | A package of community- and facility-focused educational interventions. a) Promotional educational materials. b) Educational sessions and mass media and at the community level outreach activities with give away calendars. c) Training and refresher courses were delivered to the primary health care physicians and nurses. d) Promoting culturally competent behavioural changes through messages to the targeted mothers distributed by the previously well-trained health workers. | Percent of mothers who received care during their pregnancy period. Rate of complications during pregnancy, delivery, and puerperium. Adolescent pregnancy rates. |
| Moore 2020 | Healthcare delivery behaviour (confidence, teamwork, and communication) | Government, Professional society, and international NGO | Health facility | Anaesthesia providers | Intrapartum | Education | Safer Anaesthesia from Education (SAFE) Obstetric Anaesthesia (OB) course is a three-day refresher course for trained anaesthesia providers addressing common causes of maternal mortality in LMICs. | Knowledge scores on confidence, teamwork, and communication. |
| Moshi 2021 | Healthcare seeking behaviour | Village health workers | Community | Expecting couples | Antenatal, intrapartum | Education and enablement | Community Based Continuous Training. empowerment of women on knowledge about birth preparedness and complication readiness. | Knowledge scores on birth preparedness and complication readiness. |

*(Continued)*

**Table 2.** (Continued)

| Article (First author, year) | Action (observable or measurable behaviour) | Actor (does or could do the action) | Context (location, emotional context, or social setting) | Target (person/people) | Time frame (when the action is performed) | Intervention strategy | Intervention Description | Measured outcomes |
|---|---|---|---|---|---|---|---|---|
| Oka 2022 | Self-care behaviours (Behaviours for safer childbirth) | Midwives and nurse-midwives | Health facility | Prenatal mothers | Antenatal | Training, persuasion, education, and enablement | Midwife-led prenatal group program named, "Let us chat!" (Zuinnguinza vizuri! in local language), which means a program where pregnant women communicated and connected to a midwife and pregnant women during the perinatal period. Lecture on the physiological process, danger signs, and common symptoms of pregnancy, as well as on selfcare for the symptoms of pregnancy using a movie, an audio booklet, and pictorial cards to improve knowledge of this area. An audio story booklet called Nne na Tano based on the concepts of birth preparedness and complication readiness. Sharing session provided time for sharing feelings or experiences after getting pregnant using a checklist card among pregnant women and between pregnant women and a midwife, as well as asking and discussing questions to a midwife to enhance connectedness to the midwife and peers. | Birth Preparedness and Complication Readiness knowledge scores. |
| Omer 2020 | Healthcare delivery behaviour Supporting healthy behaviours in antenatal care) | Trainers, Health information and communication programme officer, and nutrition expert. | Health facility | Healthcare providers working in ANC units | Antenatal | Education and enablement | Comprehensive in-service nutrition education and counselling training package aimed at improving counselling skills of ANC providers working in ANC units | ANC providers' engagement with the client. Nutritional messages delivered to pregnant women. |
| Onono 2021 | Healthcare-seeking behaviour (Adherence and retention) | Community mentor mothers (CMMs) | Health facility and Community | Pregnant women on ART | Antenatal, intrapartum, postpartum | Modelling, persuasion, education, and training | Community mentor mothers and text Messaging. Mobile phone text messages involved bidirectional communication between providers and mothers and encouraged uptake of important maternal and child health behaviours and services (good nutrition, antenatal clinic attendance, birth planning and skilled delivery) and engagement and adherence to HIV care. | Preterm delivery, low birth weight (LBW), miscarriage or stillbirth. |
| Oosthuizen 2019 | Healthcare delivery behaviour (Respectful delivery of obstetric care) | District clinical specialist team (multi-professional healthcare workers) | Health facility | Midwives | Intrapartum | Training and environmental restructuring | Implementation of a package of interventions under the acronym CLEVER. Clinical care, Labour ward management, eliminate barriers, Verify care, Emergency obstetric simulation training, and Respectful care. | Perinatal outcomes, namely fresh stillbirths, meconium aspiration and intrapartum-related respiratory depression rates. |
| Ratcliffe 2016 | Healthcare delivery behaviour (Respectful maternity care) | Multiple stakeholders | Health facility | Multiple stakeholders | Antenatal, intrapartum, postnatal | Training and education | Participatory intervention: Open Birth Days and a Respectful Maternity Care Workshop. OBD consisted of a participatory health education session and a tour of the hospital. (Open Birth Days (OBD), a birth preparedness and antenatal care education program, and (2) a workshop for healthcare providers based on the Health Workers for Change curriculum | Patient and provider knowledge. Patient and provider attitudes and perceptions. Patient–provider communication. Patient empowerment Provider job satisfaction. Patient -provider interaction. Patient satisfaction and perceptions of quality. |

(*Continued*)

**Table 2.** (Continued)

| Article (First author, year) | Action (observable or measurable behaviour) | Actor (does or could do the action) | Context (location, emotional context, or social setting) | Target (person/ people) | Time frame (when the action is performed) | Intervention strategy | Intervention Description | Measured outcomes |
|---|---|---|---|---|---|---|---|---|
| Saaka 2017 | Healthcare-seeking behaviour | Council of Champions (CoCs), made up of key community leaders | Community | Women of reproductive age | Antenatal, intrapartum, postnatal | Education, enablement, training, and persuasion | Social and behaviour change communication (SBCC) through empowered community leaders to improve uptake of essential maternal and newborn care (MNC) services. The innovative intervention in this study was based on evidence which suggests that social norms and practices can be changed through the social and behaviour change communication (SBCC) which focuses on the community as the unit of change. On the basis of this, key community leaders (KCLs) including chiefs, queen mothers, religious leaders and traditional healers were mobilized to constitute what we termed 'Council of Champions' (CoCs) who assisted in delivering messages designed to support families in modifying high risk practices and delivered through interpersonal interactions involving reasoning and negotiation between frontline healthcare workers and target populations at the household and community level. The CoCs comprised 5–7 most influential community members who were trained and regularly supervised by the project. | Proportion of institutional deliveries. Maternal knowledge in obstetric danger signs. Adequacy of prenatal care was measured using the Adequacy of Prenatal Care Utilization Index (APNCU). Essential newborn care practices (safe cord care, optimal thermal care, good neonatal feeding practices). Negative maternal and newborn care health (MNCH) rituals and beliefs |
| Sabin 2022 | Self-care behaviour (Adherence) | Trained clinic counsellor | Health facility and Community | Pregnant women on ART | Antenatal, postnatal | Environmental restructuring, persuasion | A text reminder to receive on their cell phones if the Real-time wireless pill monitors (WPM) was unopened within two hours of the prescribed dose time. The text messages were designed to be friendly, non-stigmatizing, and not harmful (for example, using language disclosing the woman's HIV status). Examples of messages participants could choose included "Time for prayers" or "Hello, it's time." In addition to the reminders, intervention participants were eligible to receive WPM data-informed adherence and retention counselling at monthly clinic visit. | Proportion of participants achieving ≥95% adherence during the final 30 days of the intervention period. Secondary adherence outcomes encompassed ≥95% adherence over the entire intervention period and pre- and post-delivery periods. |
| Shiferaw 2016 | Healthcare-seeking behaviours (Recognising danger signs and seeking care) | Nurses and health officers | Health facility and Community | Pregnant women and healthcare workers | Antenatal, postnatal | Environmental restructuring education and persuasion | Health workers in the intervention group received an android phone (3 phones per facility) loaded with an application that sends reminders for scheduled visits during antenatal care (ANC), delivery and postnatal care (PNC), and educational messages on dangers signs and common complaints during pregnancy. | Percentage of women who had at least 4 ANC visits, institutional delivery, and PNC visits at the health centre after 12 months of implementation of the intervention. |
| Smith 2022 | Healthcare delivery behaviours (Respectful maternity care) | Multiple stakeholders | Health facility | Multiple stakeholders | Antenatal, intrapartum, postnatal | Environmental restructuring and incentivisation | Package of interventions: BETTER pain management toolkit, feedback box, provider–client promise, fresh start funds | To assess our primary outcome, "Providers give better care to clients" (1) Whether the client reports they experienced any instance of disrespect and abuse and (2) provider reports that colleagues believe that yelling at or scolding a patient is never acceptable. Second primary outcome, "Clients are more satisfied with care.", (1) client rated care as very good or excellent and (2) client reports that the provider treated me [the client] well during labour and delivery. |

*(Continued)*

**Table 2.** (Continued)

| Article (First author, year) | Action (observable or measurable behaviour) | Actor (does or could do the action) | Context (location, emotional context, or social setting) | Target (person/ people) | Time frame (when the action is performed) | Intervention strategy | Intervention Description | Measured outcomes |
|---|---|---|---|---|---|---|---|---|
| Somji 2022 | Healthcare delivery behaviour (Pregnancy-related healthy behaviours) | Nurses, Midwives, community health volunteers | Health facility and community | Pregnant women organised into groups (Lea mimba clubs) | Antenatal, intrapartum, postnatal | Training, education, and modelling | Group antenatal care (GANC) model (Lea Mimba Pregnancy Clubs). Eight to ten women of similar gestational age met with the same health provider during sessions based on the WHO-recommended eight-visit model [10] and national standards. Women and health providers discussed a range of health topics including recognition of danger signs, care of the newborn, family planning, among others. Sessions supported interactive learning and enabled discussion of challenges and problem-solving with peers. Rituals, such as opening and closing activities and singing, were used to create a sense of membership and solidarity. Women were paired to take measurements, such as weight and blood pressure and to remind each other for future appointments, to strengthen their feelings of empowerment and solidarity. Nurses and midwives received training and mentorship on the new model of care. | Knowledge ANC experience of care Empowerment Adoption of healthy behaviours |
| Tomedi 2015 | Healthcare seeking behaviours (Medical facility/SBA-seeking behaviour for delivery) | Traditional birth attendants | Community | Pregnant women | Antenatal, intrapartum | Education, and incentivisation | TBAs were recruited to educate pregnant women about the importance of delivering in healthcare facilities and were offered a stipend for every pregnant woman whom they brought to the healthcare facility. | Percentage of Skilled Birth Attendant (SBA) deliveries |
| Turan 2018 | Healthcare delivery behaviour (Perinatal health behaviours) | Trained lay health workers | Community | Pregnant women and their partners | Antenatal, postnatal | Education, training, and persuasion | Home-based couple visits delivered by lay health workers, one male and one female, trained in couple counselling, including couple HIV testing and counselling (CHTC); maternal, child, and family health information; building couple relationship skills; and linkage to facility-based HIV prevention and treatment services. | Couple HIV Testing and Counselling (CHTC) uptake, use of Maternal and Child Health Services (ANC visits, health facility delivery, postnatal checkup for infant, postpartum checkup for woman), PMTCT behaviours for HIV-positive women (exclusive breastfeeding, ART use), and infant testing uptake for HIV-exposed infants. |
| Villadsen 2015 | Healthcare delivery behaviour (Healthy behaviours during pregnancy and after delivery) | Healthcare workers | Health facilities | Pregnant women | Antenatal, postnatal | Environmental restructuring, education, and training | Trainings, supervisions, equipment, development of health education material, and adaption of guidelines. | Proximal outcomes: Improved content of care (physical examinations, laboratory testing, tetanus toxoid (TT)-immunization, health education, conduct of health professionals, and waiting time). Distal outcomes: Increased quality of care (better identification of health problems and increased overall user satisfaction with ANC). |
| Villadsen 2016 | healthcare delivery behaviour (Health behaviours and care-seeking) | ANC attendants, traditional birth attendants | Health facilities | Pregnant and postpartum women | Antenatal, postnatal | Environmental restructuring, education, and training | intervention was designed participatorily and comprised trainings, supervisions, equipment, health education material, and adaption of guidelines. | Self- reported experience with ANC and their health behaviours. |

*(Continued)*

**Table 2.** (Continued)

| Article (First author, year) | Action (observable or measurable behaviour) | Actor (does or could do the action) | Context (location, emotional context, or social setting) | Target (person/people) | Time frame (when the action is performed) | Intervention strategy | Intervention Description | Measured outcomes |
|---|---|---|---|---|---|---|---|---|
| Zamawe 2016 | Healthcare seeking behaviour (Maternal health behaviours) | Trained local research assistants | Community | Women of reproductive age | Antenatal, postnatal | Education, modelling, and persuasion | Phukusi la Moyo (PLM) campaign—a participatory radio campaign designed to provide comprehensive maternal health information to women of reproductive age (15–49 years). The PLM's emphasis was on raising awareness of the risks associated with pregnancy, the importance of antenatal and postnatal care, and the advantages of delivering at the health facility. The campaign also aimed to promote (a) men's engagement in maternal health, (b) the use of mosquito bed-nets and (c) uptake of malaria prophylaxis among pregnant women. The PLM campaign messages were delivered by means of different activities. These included panel discussions, community discussions, drama, and songs. Members of the women's groups developed the radio programs with the technical support of the MaiMwana project and the Mudziwanthu radio station. The communities formed radio listening clubs, where women could listen to the program in groups and discuss the key messages afterwards. | Utilisation of antenatal care services. Use of mosquito bed nets during pregnancy. Delivery at the health facility. Utilisation of postnatal care services. Use of contraceptives. |

**Table 3. Behaviour change strategies.**

| Article | Number of intervention functions | Behaviour change intervention strategy | | | | | | | | |
|---|---|---|---|---|---|---|---|---|---|---|
| | | Education | Persuasion | Incentivisation | Coercion | Training | Enablement | Modelling | Environmental restructuring | Restrictions |
| Abuogi 2022 | 3 | √ | √ | | | | √ | | | |
| Abuya 2015 | 2 | | √ | | | √ | | | | |
| Ahrari 2006 | 3 | √ | √ | | | | √ | | | |
| Akor 2019 | 2 | | | | | √ | √ | | | |
| Balami 2019 | 1 | √ | | | | | | | | |
| Balami 2021 | 1 | √ | | | | | | | | |
| Bolan 2018 | 2 | √ | | | | √ | | | | |
| Cao 2021 | 2 | √ | | | | | √ | | | |
| Cohen 2017 | 1 | | | √ | | | | | | |
| Demilew 2020 | 3 | √ | √ | | | | √ | | | |
| deVries 2015 | 2 | √ | | | | √ | | | | |
| Diddana 2018 | 1 | √ | | | | | | | | |
| Doyle 2018 | 4 | √ | √ | | | √ | | √ | | |
| Everett-Murphy 2010 | 3 | √ | √ | | | | √ | | | |
| Fayorsey 2019 | 2 | √ | √ | | | | | | | |
| Futterman 2010 | 2 | √ | | | | √ | | | | |
| Hackett 2018 | 2 | | √ | | | | √ | | | |
| Jones 2013 | 2 | | | | | √ | √ | | | |
| Jones 2020 | 2 | √ | √ | | | | | | | |
| Kamau 2019 | 1 | √ | | | | | | | | |
| Katenga-Kaunda 2021 | 3 | √ | | | | | √ | √ | | |
| Kim 2021 | 2 | √ | √ | | | | | | | |
| Kinuthia 2021 | 1 | | √ | | | | | | | |
| Kujawski 2017 | 1 | | | | | | | | √ | |
| Lau 2014 | 1 | √ | | | | | | | | |
| Lewycka 2013 | 3 | √ | | | | | √ | √ | | |
| Lori 2017 | 1 | √ | | | | | | | | |
| Lund 2012 | 3 | √ | √ | | | | | | √ | |
| Maldonado 2020 | 2 | √ | √ | | | | | | | |
| Maldonado 2020 | 3 | | | √ | | √ | √ | | | |
| Matseke 2013 | 1 | | | | | √ | | | | |
| McConnell 2016 | 1 | | | | | √ | | | | |
| McConnell 2018 | 1 | | | √ | | | | | | |
| Merriel 2021 | 1 | | | | | | √ | | | |
| Metwally 2020 | 1 | √ | | | | | | | | |
| Moore 2020 | 1 | √ | | | | | | | | |
| Moshi 2021 | 2 | √ | | | | | √ | | | |
| Oka 2022 | 4 | √ | √ | | | √ | √ | | | |
| Omer 2020 | 2 | √ | | | | | √ | | | |

*(Continued)*

**Table 3.** (Continued)

| Article | Number of intervention functions | Behaviour change intervention strategy | | | | | | | | |
|---|---|---|---|---|---|---|---|---|---|---|
| | | Education | Persuasion | Incentivisation | Coercion | Training | Enablement | Modelling | Environmental restructuring | Restrictions |
| Onono 2021 | 4 | √ | √ | | | √ | | √ | | |
| Oosthuizen 2019 | 2 | | | | | √ | | | √ | |
| Ratcliffe 2016 | 2 | √ | | | | √ | | | | |
| Saaka 2017 | 4 | √ | √ | | | √ | √ | | | |
| Sabin 2022 | 2 | | √ | | | | | | √ | |
| Shiferaw 2016 | 3 | √ | √ | | | | | | √ | |
| Smith 2022 | 2 | | | √ | | | | | √ | |
| Somji 2022 | 3 | √ | | | | √ | | √ | | |
| Tomedi 2015 | 2 | √ | | √ | | | | | | |
| Turan 2018 | 3 | √ | √ | | | √ | | | | |
| Villadsen 2015 | 3 | √ | | | | √ | | | √ | |
| Villadsen 2016 | 3 | √ | | | | √ | | | √ | |
| Zamawe 2016 | 3 | √ | √ | | | | | √ | | |
| Total Count = 111 | | 37 | 20 | 5 | 0 | 19 | 16 | 6 | 8 | 0 |

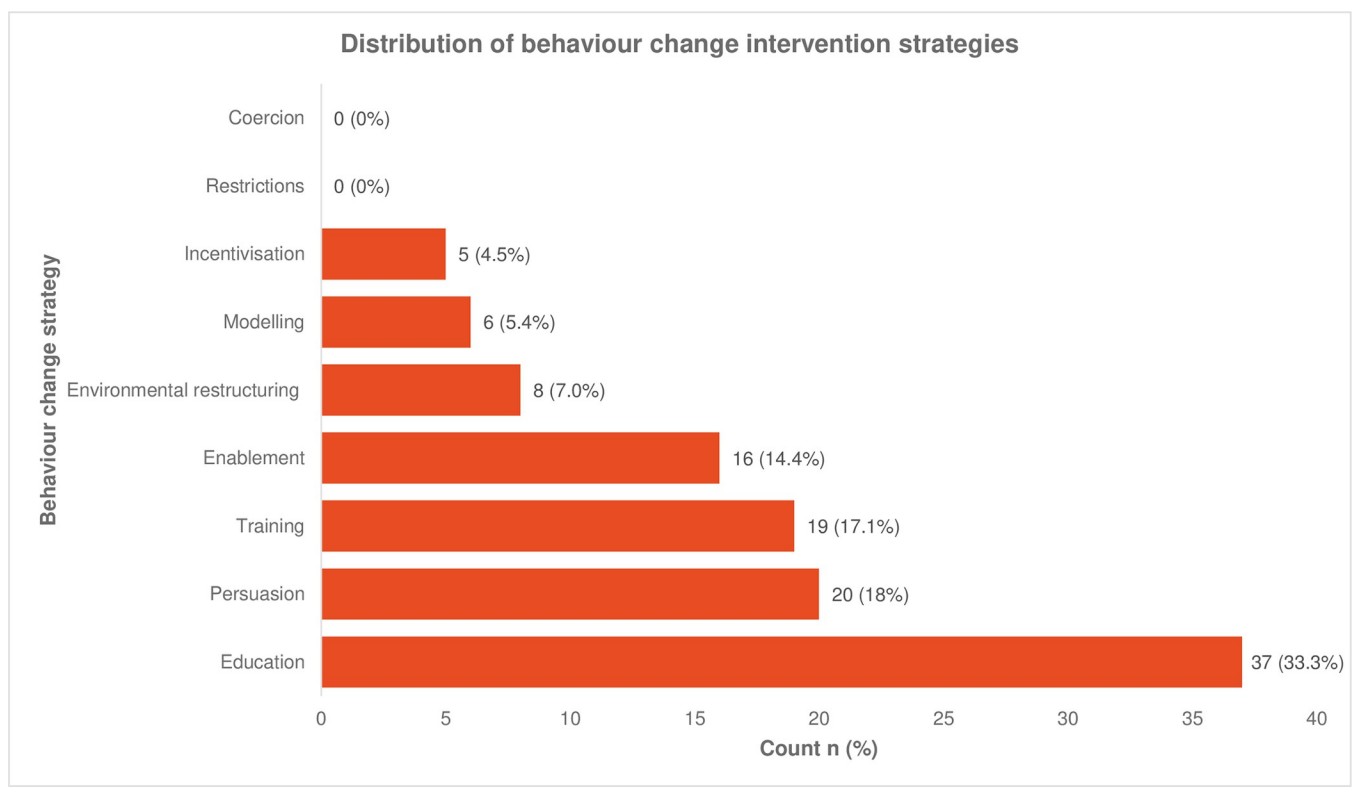

**Fig 4. Distribution of behaviour change intervention strategies.** Presented as counts and percentages. The total count from all 52 studies = 111.

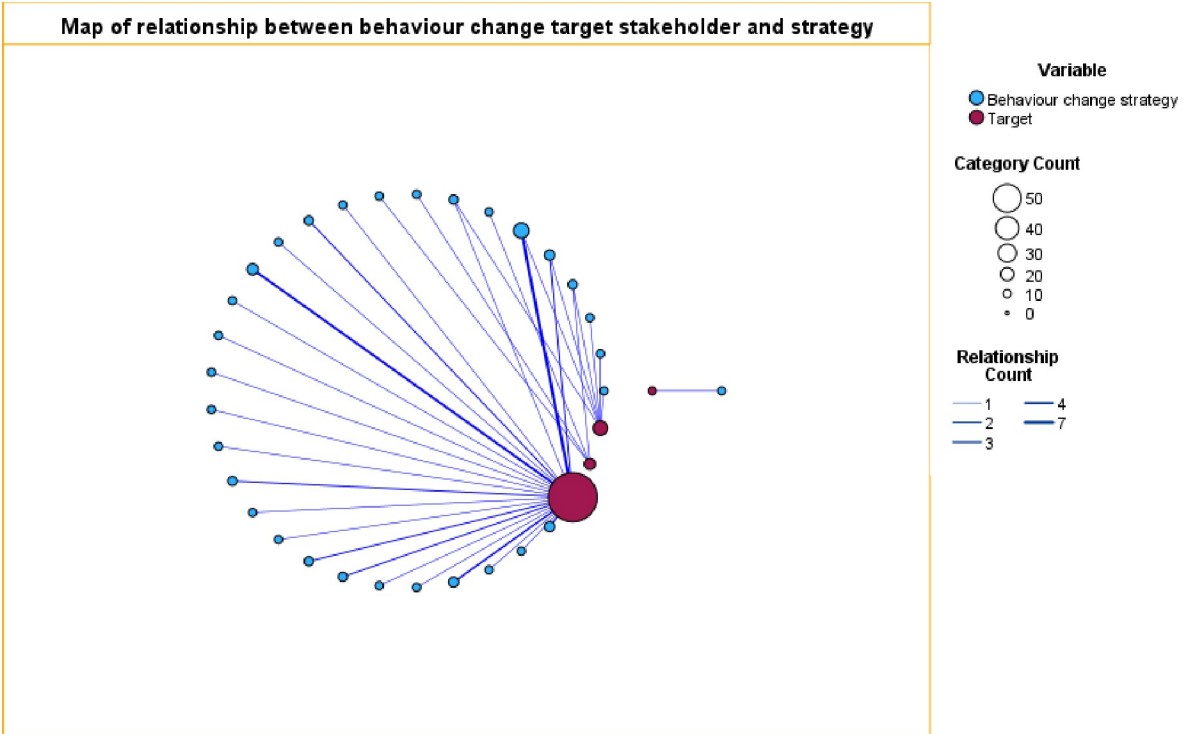

**Fig 5. Behaviour change strategies relationship map.** Those accessing maternal healthcare (biggest circle), those delivering maternal healthcare (medium circle) and mixed targets (smallest circle). The thickened lines illustrate more than one strategy. Various behaviour change strategies, exclusively or in combination, were utilised with some overlap.

of mother-to-child transmission of HIV [69] and tackling alcohol abuse in pregnancy [70]. It is not clear why these interventions were ineffective.

## Discussion

According to our knowledge, this is the only systematic review that broadly explores and presents an overview of the landscape of behavioural interventions for improving maternal health outcomes in sub-Saharan Africa. Our objective was to extract and synthesise the target behaviours, component behaviour change strategies and outcomes of behaviour change interventions for improving maternal health outcomes in sub-Saharan Africa.

We found mixed-quality evidence from randomised trials and quasi-experimental studies conducted in 12 sub-Saharan Africa countries: Kenya, Ethiopia, South Africa, Malawi, Ghana, Nigeria, Egypt, Zambia, Uganda, Rwanda, the Democratic Republic of Congo, and the United Republic of Tanzania. Using the Joanna Briggs tools to assess the quality of randomised trials and quasi-experimental studies, we found that most included studies had either a high (30 studies) or uncertain (12 studies) risk of bias. Only ten studies had a low risk of bias.

Scholars have reported in reviews that behavioural studies carry a high risk of bias, and it remains unclear whether this results from methodological flaws, reporting issues, or a mismatch of existing quality assessment tools to behavioural studies [85]. Tools such as the risk of bias justification table (RATIONALE) were designed to improve the quality of the conduct, reporting and assessment of behavioural trials [86]. It is crucial to consider our systematic review findings in light of the risk of bias categories we reported and the tool we utilised for quality assessment.

A greater proportion of studies (79%) aimed to change behaviour amongst those accessing maternal healthcare services than those delivering maternal healthcare services (13%). The interventions utilised various behaviour change strategies, the most common being education for those accessing maternal healthcare services and training for those delivering maternal healthcare services. Most of the behaviour change interventions were reported as effective.

So far, behaviour change interventions have focused mainly on those accessing maternal health services compared to those responsible for delivering the services within the health facilities. This skewed focus can partly be explained by the years of investment in efforts to encourage women to access maternal healthcare services and deliver under-skilled birth attendance [87]. However, despite more women coming to deliver within healthcare facilities, there has been concern regarding the quality of healthcare delivery contributing to suboptimal outcomes, including avoidable maternal deaths [88].

Multiple factors, such as suboptimal service delivery, contribute to suboptimal maternal health outcomes, with some being due to the attitude and behaviours of those delivering maternal health care [10]. Indeed, systematic review evidence published in 2015 documented a broad range of negative attitudes and behaviours by those delivering maternal healthcare that affect patient well-being, care satisfaction and care seeking [9].

Education of those accessing maternal health services and training those delivering maternal healthcare were the leading behaviour change strategies in our systematic review. This finding was similar to a scoping review on approaches for changing behaviour in pregnant women that found the educational approach to be the dominant strategy [89].

Quite often, education and training were used in combination with other strategies. Combining intervention strategies is a common practice, as reported in a 2017 Cochrane review on psychosocial interventions for supporting women to stop smoking during pregnancy [90]. Among healthcare workers, an overview of systematic reviews published in 2015 found that bundles of professional behaviour change interventions in healthcare seemed more effective when packaged together than as a single intervention [91].

The finding of training as the dominant intervention strategy, compared to other behaviour change intervention strategies, can be partially elucidated by the "failure to rescue" concept. The concept describes a failure or delay in recognizing and responding to patient deterioration, leading to morbidity or mortality [92].

In an attempt to mitigate against failure to rescue, it is assumed that a lack of or outdated training is the cause. As a result, the training of healthcare workers is often prioritised over other elements in the healthcare system, such as organisational, environmental, people, tasks, technology and tools factors that may equally impact healthcare delivery and outcomes [93, 94].

Although most behavioural interventions were effective, a few had either ineffective or equivocal outcomes. We did not determine the reasons for the ineffective or equivocal outcomes. However, evidence from the field of behavioural science suggests that the outcome of a behaviour change intervention is influenced by a variety of methodological and contextual factors [14, 95]. We postulate that methodological flaws and contextual factors may explain the failure of behavioural interventions to achieve the desired outcomes.

## Interpretation of the findings

There is evidence of using a range of behaviour change approaches to improve maternal health outcomes in some countries in sub-Saharan Africa. A greater proportion of previous interventions targeted those accessing maternal healthcare services compared to those delivering the services. This is important because current evidence suggests that the quality of care within

healthcare facilities ought to be a focus of initiatives aimed at improving maternal health outcomes, including tackling avoidable maternal deaths.

Regarding the scope of focus: Existing behavioural interventions focus on tackling challenges relating to human factors in service delivery, such as respectful care and teamwork. None of the studies focused on behavioural change aspects of managing specific obstetric conditions, such as obstetric haemorrhage, sepsis, hypertensive disorders, unsafe abortion, and obstructed labour, which are the leading causes of direct maternal deaths in sub-Saharan Africa.

Finally, although some behaviour change interventions were successful, it is essential to consider other context-specific factors that may influence external validity and scalability at other health facilities, regions, or countries.

## Strengths and limitations

The main strength of our systematic review is the novel use of innovative behavioural science frameworks (AACTT framework and BCW) to synthesise our findings with the effect of providing depth and clarity in categorising behaviours and intervention strategies. In addition, our systematic review provides a comprehensive overview of the behavioural interventions landscape in maternal health and the sub-Saharan Africa context. The comprehensive scope was aided by a robust search strategy and a wide variety of article sources focussing on sub-Saharan Africa, including from Africa-focused databases. The PRISMA (Preferred Reporting Items for Systematic Reviews and Meta-Analyses) guidelines were followed in our systematic review to ensure consistency in reporting. The PRISMA checklist is presented in **S3 Table**.

Our systematic review has several limitations. First, most included studies exhibited a high or uncertain risk of bias, potentially impacting the reliability of findings. This finding of moderate to high risk of bias is not unexpected in behaviour change interventional studies [85]. Behavioural scientists acknowledge biases that may affect behaviour change intervention studies, such as performance and contamination bias, and advise adopting methodological strategies to mitigate against biases [86]. Second, our choice of a narrative over quantitative synthesis meant that we could not establish the effect size of the strength of the effectiveness of the interventions. Finally, while we found evidence of effective behaviour change interventions, which could indirectly lead to a decrease in avoidable maternal deaths, a limitation is that none of the interventions directly correlated behaviour change with a reduction in the maternal mortality rate.

## Implications for maternal healthcare workers, policymakers, and governments in sub-Saharan Africa

There is a need to recognise the currently underused potential of behaviour change approaches to improve maternal health outcomes. Despite shared challenges in tackling maternal health problems, some health facilities, regions, or countries may perform better than others due to unique, effective behaviours and practices.

If identified, these unique practices and behaviours can then be decentralised from the high-performing health facilities through knowledge sharing, leading to positive change in outcomes in the poorly performing health facilities. The overall impact of embracing behavioural interventions in conjunction with existing strategies may be an improvement in maternal health outcomes and a faster reduction in the burden of avoidable maternal deaths in sub-Saharan Africa.

## Suggestions for future research

Our systematic review identified research and practice gaps which could inform future research. First, the high number of articles with a high and uncertain risk of bias indicates a

need to conduct further research towards improving behaviour change research methodology. Second, behaviour change implementation researchers at health facilities or regions with high maternal mortality should consider assessing the direct impact of the behaviour change intervention on maternal mortality rate. We acknowledge that this may be a challenge to investigate at health facilities or regions where maternal deaths are a rare occurrence. Third, as most interventions have been pre-hospital and involving those accessing healthcare, there is a need for future research to explore targeting behaviours and their influencing factors related to maternal health outcomes at the health facility level. Specifically, the identification of the challenges that healthcare workers face and how they modify their practices and behaviour to achieve better maternal health outcomes. If unique mitigation practices and behaviours are identified at a health facility, they could be explored for the feasibility of solving similar challenges nearby health facilities face, therefore, improving maternal health outcomes by overcoming local challenges with local shared solutions.

## Conclusion

Although more research is required to improve the scope and methodological quality of the current evidence base, behaviour change interventions targeted at those accessing and/or delivering maternal healthcare exist and may have a role in improving maternal health outcomes and tackling avoidable maternal deaths in Sub-Saharan Africa. They should be reviewed for incorporation into existing strategies for improving maternal health outcomes and tackling the causes of avoidable maternal deaths.

There is evidence of effective behaviour change interventions targeted at those accessing and/or delivering maternal healthcare in sub-Saharan Africa. However, more focus should be placed on behaviour change by those delivering maternal healthcare within the health facilities to fast-track the reduction of the huge burden of avoidable maternal deaths in sub-Saharan Africa.

## Supporting information

**S1 Data. Included studies and extracted data.**
(XLSX)

**S2 Data. Framework analysis and coding.**
(XLSX)

**S3 Data. ROBVIS generic dataset.**
(XLSX)

**S1 Table. Search strategy and output.**
(DOCX)

**S2 Table. Quality assessment.**
(XLSX)

**S3 Table. PRISMA checklist.**
(PDF)

## Acknowledgments

Library and Knowledge Service, Birmingham Women's and Children's NHS Foundation Trust and University Librarians, University of Birmingham.

## Author Contributions

**Conceptualization:** Francis G. Muriithi.

**Data curation:** Francis G. Muriithi, Ruth W. Gakuo.

**Formal analysis:** Francis G. Muriithi, Fabiana Lorencatto.

**Funding acquisition:** Arri Coomarasamy.

**Investigation:** Francis G. Muriithi, Fabiana Lorencatto.

**Methodology:** Francis G. Muriithi, Aduragbemi Banke-Thomas, Gillian Forbes, Eleanor Thomas, Ioannis D. Gallos, Fabiana Lorencatto.

**Project administration:** Francis G. Muriithi.

**Resources:** Arri Coomarasamy, Fabiana Lorencatto.

**Supervision:** Ioannis D. Gallos, Adam Devall, Arri Coomarasamy, Fabiana Lorencatto.

**Validation:** Gillian Forbes, Fabiana Lorencatto.

**Visualization:** Francis G. Muriithi.

**Writing – original draft:** Francis G. Muriithi.

**Writing – review & editing:** Francis G. Muriithi, Aduragbemi Banke-Thomas, Gillian Forbes, Ruth W. Gakuo, Eleanor Thomas, Ioannis D. Gallos, Adam Devall, Arri Coomarasamy, Fabiana Lorencatto.

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
