## [Decision Letter · Decision Letter 0]

8 Aug 2023

PGPH-D-23-00618

Behavioural interventions for improving maternal health outcomes in sub-Saharan Africa: A systematic review

Dear Dr. Muriithi,

Thank you for submitting your manuscript to PLOS Global Public Health. After careful consideration, we feel that it has merit but does not fully meet PLOS Global Public Health’s publication criteria as it currently stands. Therefore, we invite you to submit a revised version of the manuscript that addresses the points raised during the review process.

We look forward to receiving your revised manuscript.

Kind regards,

Shifa S. Habib

Academic Editor

Journal Requirements:

1. e noticed you have some minor occurrence of overlapping text with the following previous publication(s), which needs to be addressed:

- https://journals.plos.org/globalpublichealth/article?id=10.1371%2Fjournal.pgph.0000385

In your revision ensure you cite all your sources (including your own works), and quote or rephrase any duplicated text outside the methods section. Further consideration is dependent on these concerns being addressed.

Reviewers' comments:

Reviewer's Responses to Questions

**Comments to the Author**

1. Does this manuscript meet PLOS Global Public Health’s publication criteria? Is the manuscript technically sound, and do the data support the conclusions? The manuscript must describe methodologically and ethically rigorous research with conclusions that are appropriately drawn based on the data presented.

Reviewer #1: Yes

Reviewer #2: Yes

2. Has the statistical analysis been performed appropriately and rigorously?

Reviewer #1: Yes

Reviewer #2: N/A

3. Have the authors made all data underlying the findings in their manuscript fully available (please refer to the Data Availability Statement at the start of the manuscript PDF file)?

Reviewer #1: Yes

Reviewer #2: Yes

4. Is the manuscript presented in an intelligible fashion and written in standard English?

Reviewer #1: Yes

Reviewer #2: Yes

5. Review Comments to the Author

Reviewer #1: This is a very well written study, with a novel approach to collating behaviour change approaches to improving maternal health outcomes. The description of interventions shows the wide breadth of interventions being done across the continent, and how effective the interventions are. Many of the studies were published in the last 13 years. The fact that more interventions were directed at health seeking behaviour rather than healthcare worker behaviour and skills was also very interesting. Meaning there is greater drive towards primordial and primary prevention, over secondary and tertiary prevention. Would be interesting to know if the interventions were mostly designed by clinical or public health research groups.

1. In the discussion I would perhaps state that this is the only systematic review 'according to your knowledge'

2. Did all the studies only measures effectivessness, or were there other measures such as acceptability or outcomes. E.g. reduction in composite outcomes such as mortality or morbidity or haemorrhage or disease. In table 2 I would suggest adding a column on outcomes measured for completeness.

3. It would be interesting to know if you believe that behaviour interventions are scalable - and whether success in one regions means they could also be successful in other regions.

4. Ultimately did the interventions have a reduction on the maternal mortality rate. If we don't know, this must be included as a limitation - as the overall framing of the paper implies that the impact of behaviour change is reduction in maternal mortality. However the findings of the systematic review, overall suggest that behaviour change interventions are effective, without correlating change in behaviour with MMR.

5. In the discussion it may be useful to include the concept of 'failure to rescue' as a reason to include more training interventions.

Reviewer #2: Abstract: I suggest that the authors add information on the methodology used for the systematic review. Include data source and eligibility criteria.

Introduction: In the first paragraph mention since when the decline in preventable maternal deaths has stagnated.

Regarding the most common causes of maternal death, insert the year to which the information is related to and the rates of each cause.

I suggest that the authors structure the introduction as following:

Maternal Mortality rate

Main maternal cause of death

Interventions to reduce the MMR

Impact of behavioral interventions

Type of behavioral interventions and stakeholders influence.

Methods: review the exclusion criteria, there are not suppose to be the opposite of the inclusion criteria.

Discussion: I suggest a detailed discussion on quality of data/studies included.

6. PLOS authors have the option to publish the peer review history of their article (what does this mean?). If published, this will include your full peer review and any attached files.

**Do you want your identity to be public for this peer review?** For information about this choice, including consent withdrawal, please see our Privacy Policy.

Reviewer #1: No

Reviewer #2: No

---

## [Decision Letter · Decision Letter 1]

12 Dec 2023

PGPH-D-23-00618R1

Behavioural interventions for improving maternal health outcomes in sub-Saharan Africa: A systematic review

Dear Dr. Muriithi,

Thank you for submitting your manuscript to PLOS Global Public Health. After careful consideration, we feel that it has merit but can improve further to fully meet PLOS Global Public Health’s publication criteria as it currently stands. Therefore, we invite you to submit a revised version of the manuscript that addresses the points raised during the review process.

We look forward to receiving your revised manuscript.

Kind regards,

Shifa S. Habib

Academic Editor

Journal Requirements:

1. We noticed you have some minor occurrence of overlapping text with the following previous publication(s), which needs to be addressed:

- https://journals.plos.org/globalpublichealth/article?id=10.1371%2Fjournal.pgph.0000385

In your revision ensure you cite all your sources (including your own works), and quote or rephrase any duplicated text outside the methods section. Further consideration is dependent on these concerns being addressed.

Additional Editor Comments (if provided):

Reviewers' comments:

Reviewer's Responses to Questions

**Comments to the Author**

1. If the authors have adequately addressed your comments raised in a previous round of review and you feel that this manuscript is now acceptable for publication, you may indicate that here to bypass the “Comments to the Author” section, enter your conflict of interest statement in the “Confidential to Editor” section, and submit your "Accept" recommendation.

Reviewer #2: (No Response)

Reviewer #3: (No Response)

2. Does this manuscript meet PLOS Global Public Health’s publication criteria? Is the manuscript technically sound, and do the data support the conclusions? The manuscript must describe methodologically and ethically rigorous research with conclusions that are appropriately drawn based on the data presented.

Reviewer #2: Yes

Reviewer #3: Yes

3. Has the statistical analysis been performed appropriately and rigorously?

Reviewer #2: N/A

Reviewer #3: No

4. Have the authors made all data underlying the findings in their manuscript fully available (please refer to the Data Availability Statement at the start of the manuscript PDF file)?

Reviewer #2: Yes

Reviewer #3: Yes

5. Is the manuscript presented in an intelligible fashion and written in standard English?

Reviewer #2: Yes

Reviewer #3: Yes

6. Review Comments to the Author

Reviewer #2: Please revise the exclusion criteria, those presented are the opposite of the inclusion criteria, this is not the definition of an exclusion criteria. Please read about eligibility (inclusion and exclusion criterias), to know how to formulate them and edit accordingly.

Reviewer #3: Title: A Systematic Review of Behavior Change Interventions to Improve Maternal Health Outcomes in Sub-Saharan Africa

Strengths:

Innovative Frameworks: The utilization of the AACTT framework and BCW adds depth and clarity to the review, providing a solid foundation for categorizing behaviors and intervention strategies.

Comprehensive Scope: The review covers a wide range of studies from various Sub-Saharan African countries, offering a comprehensive overview of the behavioral landscape related to maternal health.

Limitations:

Risk of Bias: The majority of included studies exhibited a high or uncertain risk of bias, potentially impacting the reliability of findings. The review acknowledges this limitation but does not delve deeply into its implications.

Outcome Measurement: While effective interventions are identified, the review falls short of directly correlating behavior change with a reduction in the maternal mortality rate. Future research might benefit from exploring this link more explicitly.

7. PLOS authors have the option to publish the peer review history of their article (what does this mean?). If published, this will include your full peer review and any attached files.

**Do you want your identity to be public for this peer review?** For information about this choice, including consent withdrawal, please see our Privacy Policy.

Reviewer #2: No

Reviewer #3: **Yes: **Amanuel Abajobir, PhD

---

## [Editor Report · Decision Letter 2]

31 Jan 2024

A Systematic Review of Behaviour Change Interventions to Improve Maternal Health Outcomes in sub-Saharan Africa.

PGPH-D-23-00618R2

Dear Muriithi,

We are pleased to inform you that your manuscript 'A Systematic Review of Behaviour Change Interventions to Improve Maternal Health Outcomes in sub-Saharan Africa.' has been provisionally accepted for publication in PLOS Global Public Health.

Best regards,

Shifa S. Habib

Academic Editor
